# Fermented Plant-Based Milks Based on Chestnut and Soybean: Comprehensive Evaluation of Fermentation Characteristics and Aroma Profiles Using Four Lactic Acid Bacteria Strains

**DOI:** 10.3390/foods14142511

**Published:** 2025-07-17

**Authors:** Qingyang Sun, Xiaowen Shi, Yue Zhao, Ruiguo Cui, Yaya Yao, Xiaoyu Liu, Haoran Wang, Li Zhang, Lijun Song

**Affiliations:** 1College of Food Science and Technology, Hebei Normal University of Science and Technology, Qinhuangdao 066600, China; sunqingyang0913@163.com (Q.S.); sxw0131@126.com (X.S.); zhaoyue_0335@126.com (Y.Z.); ruigc_hnust@163.com (R.C.); yaoyaya6694@163.com (Y.Y.); liuxiaoyu202202@126.com (X.L.); rannar1126@163.com (H.W.); cxbh1984@163.com (L.Z.); 2Chestnut Research Center, Hebei Normal University of Science and Technology, Qinhuangdao 066004, China; 3Hebei Fruit Processing Technology Innovation Center, Hebei Normal University of Science and Technology, Qinhuangdao 066600, China; 4Hebei Key Laboratory of Natural Products Activity Components and Function, Hebei Normal University of Science and Technology, Qinhuangdao 066004, China

**Keywords:** chestnut, plant-based milk, lactic acid bacteria, fermentative characteristics, flavor

## Abstract

In this study, four lactic acid bacteria (LAB) strains, including *Lactiplantibacillus plantarum* CICC21790, *Lacticaseibacillus casei* CICC6117, *Lacticaseibacillus rhamnosus* ATCC7469, and *Limosilactobacillus fermentum* CICC22704, were used to ferment a plant-based milk composed of chestnut and soybean. The fermentative characteristics of the four LAB strains and the aroma characteristics of the resulting plant-based milks were systematically investigated. The results showed that all four LAB strains successfully fermented the plant-based milk. The viable cell counts ranged from 7.67 to 8.57 lg CFU mL^−1^, and pH values were between 3.80 and 4.10. Comprehensive analyses performed using E-nose and HS-GC-IMS revealed distinct aroma characteristics in plant-based milks fermented by different LAB strains. Specifically, LAB fermentation, particularly by the CICC22704, significantly reduced the concentrations of aldehydes (e.g., hexanal, heptanal), thereby diminishing Green aroma characteristics. The increased concentrations of alcohols (e.g., 1-pentanol), ketones (e.g., 2, 3-butanedione) and furan compounds (e.g., 2-pentylfuran) in fermented plant-based milks enhanced Pungent, Creamy, and Fruity aroma characteristics, respectively. Fermentation by CICC21790, ATCC7469, and CICC6117 may result in stronger intensities of these three aroma characteristics compared to fermentation by the CICC22704. For the Fatty aroma characteristic, it was enhanced by CICC21790 fermentation but diminished by ATCC7469, CICC6117, and CICC22704 fermentations.

## 1. Introduction

Nowadays, fermented plant-based milk has attracted considerable attention as an alternative to traditional dairy products due to its health benefits and sustainability. Unlike conventional dairy-based yogurt, fermented plant-based milk is produced from plant-derived raw materials through lactic acid bacteria (LAB) fermentation [1]. LAB utilizes the rich nutrients present in plant-based materials to produce organic acids, bacteriocins, and other active ingredients to enhance the antibacterial ability of fermented plant-based milk [2]. This product is particularly suitable for individuals with lactose intolerance or milk protein allergies, given its low saturated fat content, low calorie levels, and the absence of lactose [3]. Moreover, the abundant dietary fiber and diverse plant-derived bioactive compounds in fermented plant-based milk align well with consumer demand for health-promoting and environmentally sustainable food products [4].

Compared to traditional dairy products, fermented plant-based milk still exhibits certain limitations, such as suboptimal texture, relatively lower nutritional value, undesirable flavor, and coarse mouthfeel. Fermented soymilk often exhibits a heavy soya flavor despite its high protein content, essential amino acids, and B vitamins [5,6,7]. As reported previously, during soymilk fermentation, the addition of some carbohydrates (such as glucose or lactose) can modulate the glycolysis/gluconeogenesis and pyruvate metabolic pathways, thereby promoting the production of metabolites including 2, 3-butanedione, acetoin, acetaldehyde, and acetic acid [8]. These flavor substances can improve the undesirable flavor of plant-based fermented products to some extent [9]. As is well known, chestnuts are highly favored by consumers due to their unique flavor, texture and nutritional richness [10,11]. Starch constitutes the primary component of chestnuts, accounting for approximately 70–80% on a dry weight basis [12]. Following saccharification, chestnuts exhibit increased sugar content, which can serve as an energy source for the growth and metabolism of LAB [13]. Moreover, the oligosaccharides produced during the saccharification process have been shown to effectively promote the proliferation of LAB [12]. Additionally, chestnuts are rich in polysaccharides, flavonoids, polyphenols, and other bioactive substances, which confer numerous physiological benefits, including antioxidant, hypoglycemic, and anti-inflammatory properties [14]. Therefore, we attempted to blend soymilk with saccharified chestnut solution to produce a fermented plant-based milk. This approach enables us to leverage the high protein content of soymilk as well as the abundant sugar content and distinctive aroma of chestnuts.

LAB fermentation is highly effective in enhancing the organoleptic qualities of plant-based fermented milk and mitigating heavy undesirable flavors. Studies on fermented soymilk have demonstrated that fermentation with *Lactiplantibacillus plantarum* significantly reduces the grassy taste commonly associated with plant proteins [15]. Moreover, fermentation using *Streptococcus thermophilus*, *Lactiplantibacillus plantarum*, *Bifidobacterium*, and *Lacticaseibacillus casei* individually could markedly reduce the concentrations of hexanal and 1-octen-3-ol, which are derived from soymilk [16,17,18,19] and lead to the typical soybean off-flavor [9]. Additionally, LAB fermentation increases the levels of acetoin, 2, 3-butanedione, acetic acid, and other compounds that contribute to a frankincense-like aroma in fermented plant-based milk [16,17,18,19]. Zheng et al. [9] utilized *Lactobacillus harbinensis* M1, which has a high capacity for producing 2, 3-butanedione and acetoin, to enhance the creamy and fruity flavor and improve the overall flavor acceptability of fermented soymilk. Therefore, the LAB strains play a crucial role in transforming flavor compounds during fermentation, with different strains exerting varied influences on the flavor of fermented plant-based milk.

Flavor analysis of fermented plant-based milk can be conducted using a variety of methods, including sensory evaluation and instrumental techniques. Sensory evaluation relies on olfactory assessments, which demand extensive expertise and often result in subjective outcomes [20]. Therefore, instrumental techniques are commonly employed to analyze sample differences. The electronic nose (E-nose) is an analytical instrument that mimics the human olfactory system, offering faster and more accurate results compared to human perception [21]. However, since the E-nose signal output typically represents a composite response of multiple compounds, it is unable to identify key flavor compounds or evaluate their contributions to the overall aroma profile [22]. In recent years, HS-GC-IMS has been widely adopted owing to its numerous advantages, including a short analysis time, simple sample preparation, and high sensitivity to low-molecular-weight volatile compounds [23], in comparison with GC-MS. Therefore, in this study, E-nose and HS-GC-IMS were selected to analyze the differences in overall aroma profiles and volatile compound concentrations between unfermented and fermented plant-based milks.

This study aimed to evaluate the fermentative characteristics of different LAB strains, including *Lactiplantibacillus plantarum* CICC21790, *Lacticaseibacillus casei* CICC6117, *Lacticaseibacillus rhamnosus* ATCC7469, and *Limosilactobacillus fermentum* CICC22704, and aroma characteristics of fermented plant-based milks. Multiple statistical analyses were employed to identify key flavor compounds and assess their contributions to the overall aroma profile in plant-based milks. The findings will provide a theoretical basis for flavor improvement and offer new choices for the development of fermented plant-based milks.

## 2. Materials and Methods

### 2.1. Materials, Strains, and Chemical Reagents

Yanbao chestnuts (*Castanea mollissima* Blume) were harvested on 25 September 2023, from a laboratory-owned plantation in Qinhuangdao City, Hebei Province, China. After collection, the chestnuts were dehulled using a special shelling machine (Small and simple type, Jiangsu Xinda Machinery Manufacturing Co., Ltd., Nanjing, China) and then stored at −20 °C for future use. Soybeans [*Glycine max* (L.) Merrill] were purchased from the local market in Qinhuangdao, China, in 2024 and kept at 4 °C until use. The strains used in this study included *Lactiplantibacillus plantarum* CICC21790 (CICC21790), *Lacticaseibacillus casei* CICC6117 (CICC6117), *Lacticaseibacillus rhamnosus* ATCC7469 (ATCC7469), and *Limosilactobacillus fermentum* CICC22704 (CICC22704), which were purchased from the China Center of Industrial Culture Collection (Beijing, China). All strains were cultivated in Man Rogosa Sharpe (MRS) broth at 37 °C for 36 h and preserved in 25% glycerol at −80 °C for subsequent experiments. Chemicals and reagents, including DNS (3, 5-Dinitrosalicylic acid) reagent, sodium hydroxide, phenolphthalein, KHP (potassium hydrogen phthalate), MRS agar, and MRS broth, were acquired from Solarbio Co., Ltd. (Beijing, China).

### 2.2. Preparation of Plant-Based Milk

Plant-based milk used for fermentation was prepared using chestnut and soybean. Shelled raw chestnuts were thawed at room temperature, sliced, baked for 2.5 h, ground into powder, and sieved through a 100-mesh screen to obtain chestnut powder. The chestnut powder was mixed with double-distilled water at a ratio of 1:9 (m/V), gelatinized in a magnetic stirring oil bath pot (Four-hole variable temperature and variable stirring digital display model, Hunan Qianyan Technology Co., Ltd., Changsha, China) at 90 °C for 10 min, cooled to 70 °C, and subjected to liquefaction with 0.25% α-amylase for 2 h at 70 °C. Subsequently, the mixture was cooled to 60 °C and treated with 0.25% glycosylase for saccharification for 3 h at 60 °C, yielding the saccharified chestnut solution. Meanwhile, soybeans were soaked for 12 h and homogenized with double-distilled water (1:8, m/V) for 3 min, followed by filtration through a 50-mesh sieve to obtain soymilk. The soymilk was sterilized at 100 °C for 5 min. Finally, on a clean bench (OptiClean-1300, Likang Precision Technology Co., Ltd., Shanghai, China), the saccharified chestnut solution and sterilized soymilk were mixed in a 2:3 (m/m) ratio under sterile conditions to prepare the plant-based milk. It should be noted that the ratio of the saccharified chestnut solution to sterilized soymilk was determined through preliminary trials by evaluating the curdling properties and flavor of the resulting fermented plant-based milk.

### 2.3. Preparation of Fermented Plant-Based Milk

For the activation of four LAB strains, 0.45 mL of each cultured strain was aseptically transferred and inoculated into 50 mL of MRS broth, followed by incubation at 37 °C for 36 h under agitation (200 rpm). The resultant cultures were centrifuged at 5000× *g* at 4 °C for 5 min. The cell pellets were then resuspended and adjusted to a final concentration of approximately 5 × 10^8^ CFU mL^−1^. A volume of 2% (*v*/*v*) of the suspension was subsequently inoculated into plant-based milk. To obtain fermented plant-based milk, the inoculated samples were first fermented at 37 °C for 12 h and then matured at 4 °C for 12 h. The fermented plant-based milks obtained using CICC21790, CICC6117, ATCC7469, and CICC22704 strains were designated as CICC21790-M, CICC6117-M, ATCC7469-M, and CICC22704-M, respectively, while the unfermented plant-based milk was designated as CK. All samples were prepared in triplicate for analysis.

### 2.4. Determination of Fermentative Characteristics of LAB Strains

To evaluate the fermentative characteristics of the four LAB strains, viable cell counts, soluble solid content, reducing sugar content, pH value, and titratable acidity were measured. Viable cell counts of the LAB strains in fermented plant-based milks were determined using the dilution pour plate method on MRS agar [24]. Soluble solid content was measured with a refractometer (RX-5000; Atago, Tokyo, Japan), and the average of three replicates was expressed as °Brix (%). The concentration of reducing sugar was determined using the 3, 5-dinitrosalicylic acid (DNS) method [25]. The pH value was measured with a pH meter (FE28, Mettler-Toledo Instruments Co., Ltd., Shanghai, China). Titratable acidity was measured in according with China National standard GB 5009.239-2016 [26].

### 2.5. Determination of Volatile Compounds in Plant-Based Milks

#### 2.5.1. E-Nose Analysis

An E-nose (PEN 3, Airsense Analytics Co., Ltd., Schwerin, Germany) was employed to detect odors. The test conditions were adapted based on the method of Chai et al. [27]. Specifically, 2.0 g of each sample was accurately weighed and placed into a 20 mL headspace vial, and followed by equilibration at room temperature for 15 min. Each sample was analyzed in triplicate. The measurement conditions were set as follows: 5 s injection preparation time, 120 s detection time, 5 s autozero time, 200 s sensor cleaning time, and a carrier gas flow rate of 400 mL/min. A stable response curve observed between 50 and 60 s during the 120 s detection period was selected for data analysis. The aroma characteristics of each sample were described by the response values corresponding to the 10 sensors as presented in Table 1.

#### 2.5.2. HS-GC-IMS Analysis

Briefly, 2 g of sample was placed into a 20-mL SPME glass vial and spiked with 10 µL of the internal standard, 2-methyl-3-heptanone (10 mg/L). The vial was then sealed with a PTFE-silicone septum and equilibrated at 60 °C for 15 min under a rotational speed of 500 rpm. Subsequently, the syringe of the automatic headspace sampler unit (CTC Analytics AG, Zwingen, Switzerland) was inserted into the headspace of the sealed vial to extract 100 μL of the headspace gas at 85 °C. The extracted gas was then injected into the GC injection port in splitless mode to initiate the GC-IMS analysis.

The GC-IMS analysis was performed using the G.A.S. FlavourSpec GC-IMS system, which was equipped with an MXT-5 capillary column (15 m × 0.53 mm × 1.0 μm, Restek, PA, USA). Ultra-pure helium (99.999%) served as the carrier gas, and the flow rate was programmed as follows: it initially remained at 2 mL/min for 2 min, then increased linearly to 10 mL/min in 8 min, followed by a further increase linearly to 100 mL/min in 10 min, with a final holding time of 20 min. The drift tube temperature was maintained at 45 °C, and the drift gas velocity was 150 mL/min. Each sample was analyzed in triplicate.

The volatile compounds were identified by comparing their retention indices (RIs) and drift time (Dt) with those of pure standards in the NIST 20 library and the IMS Database (G.A.S., Dortmund, Germany). The RIs were calculated using n-ketones C4-C9 (Sinopharm Chemical Reagent Beijing Co., Ltd., Beijing, China) as external references. Quantification of the volatiles was performed using the internal standard method, wherein the GC peak volumes of the samples were compared to those of the internal standard. The results were expressed in μg/L.

### 2.6. Relative Odor Activity Value (ROAV) Analysis

ROAVs were analyzed to determine the contribution of each volatile compound to the overall aroma. The formula for ROAV [28] is as follows:
$$\mathrm{ROAV}i=\frac{Ci\times Tmax}{Ti\times cmax}\times 100$$

ROAV*i* is the relative odor activity value of the compound. *Ci* is the concentration of the compound, while *Ti* is the threshold of the compound in water. *Tmax*/*Cmax* represents the maximum of *Ci*/*Ti* among all compounds in this sample.

### 2.7. Statistical Analysis

The data are presented as the mean ± standard deviation from three independent experiments. One-way analysis of variance (ANOVA), followed by Duncan’s multiple range test (*p* < 0.05), was performed using IBM SPSS Statistics 19.0 software (SPSS Inc., Chicago, IL, USA) to evaluate significant differences. Principal component analysis (PCA) was carried out using Origin 2018 software (OriginLab Corporation, Northampton, MA, USA). Variable importance in projection (VIP) values were calculated using SIMCA 14.1 software (Sartorius, Göttingen, Germany). E-nose data were collected using the Winmuster software 1.6.2, and radar maps and column charts were generated using Origin 2018 software. Fingerprint plots and difference plots were created using VOCal 0.4.03 software (G.A.S., Dortmund, Germany).

## 3. Results and Discussion

### 3.1. Fermentative Characteristics Analysis of LAB Strains

The viable cell counts, soluble solid content, reducing sugar content, pH value, and titratable acidity of the unfermented and fermented plant-based milks were presented in Table 2. The results indicated that the four LAB strains successfully fermented the plant-based milk, achieving viable cell counts within the range of 7.67 to 8.57 lg CFU mL^−1^, and pH values between 3.80 and 4.10. The viable cell counts in the fermented samples exhibited the significant differences, with CICC6117-M showing the highest value, followed by ATCC7469-M, then CICC22704-M in third position, and CICC21790-M exhibiting the lowest count. In terms of pH value, CICC22704-M exhibited a significantly higher level compared to the other three samples, while no significant differences were observed among the remaining individual samples. Compared with the CK sample, the titratable acidity of the four fermented samples showed a significant increase, ranging from 82.47 °T to 87.77 °T, indicating this four LAB strains had a relatively high capacity for acid production. This phenomenon can likely be attributed to higher abundance of glycoside hydrolases that these strains possessed [29]. The titratable acidity of CICC22704-M and CICC21790-M was significantly higher than those of CICC6117-M and ATCC7469-M. Collectively, these findings suggest that the four LAB strains exhibit superior adaptability to the fermentation environment of a saccharified chestnut solution and soymilk mixture.

### 3.2. E-Nose Analysis of Plant-Based Milks

To gain an initial understanding of the differences in volatiles between unfermented and fermented plant-based milks, the E-nose was employed to evaluate the overall aroma profiles of these samples. As shown in Figure 1a, compared with the CK sample, the overall aroma profiles of the four fermented samples were significantly altered, as indicated by the variations in response values of the W1S, W2S, W5S, W1W, and W2W sensors. In Figure 1b, the highest response values of W1S and W2S were observed in CICC22704-M, suggesting that CICC22704 may possess a strong capacity to produce methylated compounds, hydrocarbons, alcohols, aldehydes, and ketones. For the W5S sensor, the highest response value was detected in ATCC7469-M, indicating the strong ability of ATCC7469 to produce nitrogen oxide compounds. Additionally, higher response values of the W1W and W2W sensors in both CICC22704-M and ATCC7469-M samples suggested that these two strains were also good producers of organic sulfide compounds and aromatic compounds. For the CICC21790-M sample, the response values of these five sensors were the lowest among the fermented samples, indicating the relatively limited ability of CICC21790 to produce volatile compounds. Compared with the other three strains, CICC6117 exhibited a medium capacity for producing volatile compounds, as evidenced by response values of these five sensors. These results can be attributed to the interactions between chemical compounds and microorganisms during fermentation, as well as the reduction to smaller molecules, leading to changes in the content of volatile compounds and aroma profiles [30].

To better discriminate the differences in overall flavor profiles between unfermented and fermented plant-based milks, PCA was conducted based on E-nose sensor response values. As shown in Figure 1c, the first two principal components accounted for 92.5% of the total variance, with PC1 contributing 74.2% and PC2 contributing 18.3%. After fermentation, the overall flavor shifted to the right, indicating that LAB fermentation had a more pronounced influence on PC1. Notably, the overall flavor profiles of CICC22704-M and ATCC7469-M changed greatly and appeared relatively distant from each other in the PCA score plots, suggesting that CICC22704 and ATCC7469 may possess a relatively strong ability to generate diverse aroma compounds. In contrast, the CICC21790-M and CICC6117-M samples remained relatively close to the CK sample, indicating these two strains have a limited ability to produce volatile compounds. To achieve an in-depth understanding of the differences in volatile compositions of unfermented and fermented plant-based milks, volatile compounds were further analyzed using HS-GC-IMS techniques.

### 3.3. HS-GC-IMS Analysis of Plant-Based Milks

#### 3.3.1. Volatiles Fingerprints of Plant-Based Milks

To comprehensively investigate the differences in volatile compounds between unfermented and fermented plant-based milks, HS-GC-IMS was utilized to analyze the volatile compounds. A flavor analyzer was used to identify the volatile compounds in these samples. Differential comparison topographic plots were obtained by using the CK sample as a reference (Figure 2a). Compared with the CK sample, the concentrations of various compounds in the fermented samples showed significant variation. Moreover, substantial differences were also observed among the samples fermented by different LAB strains.

To intuitively reflect the differences in volatile compositions among these samples, a fingerprint of volatile compounds was constructed. As shown in Figure 2b, a total of 80 peak signals were detected. Despite the high sensitivity of GC-IMS, the limitations of the available database resulted in only 74 compounds being qualitatively identified. Among these identified compounds, there were 20 dimers and one trimer, where water and hydrogen ions have the possibility of combining with one monomer and two monomers to form a dimer and a trimer when charged volatile compounds are given [31]. This phenomenon enhances the accuracy of HS-GC-IMS in the quantification of volatile compounds in samples, ensuring more reliable analytical results. Approximately four regions can be categorized to reflect the characteristics of differential volatile compounds in unfermented and fermented samples based on the color intensity of each point and the statistical analysis results of volatile compound peak intensities. Volatile compounds in the CK sample are primarily distributed across three regions: A, B, and C. In region A, five volatile compounds (1-hexanol [D], 2-pentylfuran, 1-hexanol [T], 2-heptanone [D], and (Z)-2-penten-1-ol) exhibited significantly elevated concentrations in the fermented samples. Conversely, in region C, the concentrations of 28 volatile compounds (e.g., hexanal [D], heptanal) were markedly reduced after fermentation, with the most pronounced decrease observed in the CICC22704-M samples. For example, the concentration of hexanal [D] in the CK sample was approximately 700 μg/L, whereas in the CICC21790-M, ATCC7469-M, and CICC6117-M samples it was around 270 μg/L, and in the CICC22704-M sample, it was approximately 120 μg/L. In region B, the concentrations of 10 volatile compounds (e.g., 1-pentanol, (Z)-3-hexenyl acetate [D]) displayed diverse changes, with some increasing and others decreasing depending on the LAB strains used for fermentation. Finally, in region D, 37 volatiles (e.g., 2, 3-butanedione, acetoin) were predominantly generated during LAB fermentation, contributing to the distinct differences among samples fermented by various LAB strains.

#### 3.3.2. Concentration of Volatiles in Plant-Based Milks

The identified compounds based on the retention index (RI) and drift time (Dt) are presented in Table 3. In this study, monomer, dimer, and trimer of a compound were collectively considered as a single compound for subsequent analysis. These compounds can be classified into 9 types: 11 aldehydes, 16 alcohols, 8 esters, 8 ketones, 2 acids, 4 furans, 3 sulfur compounds, 1 other compound, and 6 unknown compounds.

Aldehydes have adverse effects on the flavor of fermented products and constitute the most abundant class flavor compounds in unfermented plant-based milk (34.98%), likely originating from the raw materials and processing procedures. In this study, 11 aldehydes were identified, including 3-methylbutanal, (E)-2-pentenal, 3-methyl-2-butenal, hexanal, (E)-2-hexenal, heptanal, (E)-2-heptenal, (E, E)-2, 4-heptadienal, (E)-2-octenal, benzaldehyde, and benzeneacetaldehyde. Notably, with the exception of 3-methyl-2-butenal, the concentrations of all other aldehydes decreased significantly in the fermented plant-based samples. Similar results were observed in the fermentation of mung bean by *L. plantarum* [32] and the fermentation of sorghum by *Lactobacillus* species [33]. These results were mainly related to the hydrolysis of proteins and the metabolic pathways of amino acids [32]. Furthermore, aldehydes are inherently unstable compounds that can be easily reduced to alcohols or oxidized to acids within food matrices, particularly under microbial influence [34]. Among the 11 aldehydes detected, hexanal was found to be present at the highest concentration in the CK sample (36.07%). In the ATCC7469-M, CICC21790-M, and CICC6117-M samples, its concentration was reduced to approximately half of that observed in CK sample, whereas in the CICC22704-M sample, it was reduced to one-fifth of the original content. Numerous studies have confirmed that LAB fermentation is an effective approach to substantially decrease the hexanal concentration [16,17,18,19,35]. Regarding the lowest concentration of hexanal in the CICC22704-M sample, it may be attributed to the ability of CICC22704 to convert more hexanal into hexanoic acid via its inherent alcohol dehydrogenase during growth and metabolism. These acidic compounds can subsequently undergo esterification [36].

Alcohol compounds are present in substantial quantities in both unfermented and fermented samples. These compounds, which can be produced via lipid metabolism, amino acid metabolism, and the redox reactions of aldehydes, typically contribute flavors reminiscent of grass, fruit, flowers, and wine to the product [37]. Compared with the CK sample, a higher total concentration of alcohol compounds was detected in the four fermented samples; however, no significant differences were observed among these fermented samples. Among the 16 alcohols detected, hexanol was identified as the predominant alcohol in all samples, followed by 1-octen-3-ol and 1-pentanol. Hexanol, which exhibits a grassy flavor with a sensory threshold of approximately 500 μg/kg, is primarily formed from the 13-hydroperoxide produced through the lipoxygenase (LOX)-catalyzed oxidation of linoleic acid [38,39]. A higher hexanol content was observed in the fermented samples, suggesting that these four LAB strains may reduce the abundance of certain aldehydes via their aldehyde dehydrogenase activity [40], while simultaneously increasing the concentration of hexanol. For 1-octen-3-ol, previous studies have shown that it constitutes a relatively high proportion among the alcohols present in soymilk and is responsible for imparting a mushroom-like aroma. Its sensory threshold has been determined to be 10 μg/kg [32]. In fermented soymilk, this compound is primarily generated through the enzymatic cleavage of linoleic acid and plays a key role in contributing to the beany flavor [41]. Among the four LAB strains, CICC21790 and CICC22704 significantly reduced the concentration of 1-octen-3-ol, which is in agreement with prior reports [32]. This indicates that both CICC21790 and CICC22704 possess an enhanced ability to mitigate the “beany flavor.” Regarding 1-pentanol, earlier studies have reported that it is associated with a spicy flavor [42], and its concentration was found to exceed 200 μg/L in all samples. Notably, CICC22704 and ATCC7469 significantly reduced its concentration, while the CICC6117 and CICC21790 did not exhibit any significant effect on its concentration. The reduction in the partial alcohols may be related to their combination with acids to form esters during the later stage of fermentation [43].

Most ketones possess a fragrant odor and are mainly generated through the beta-oxidation of saturated fatty acids and the degradation of amino acids during the fermentation process [44]. The highest total ketones content was observed in the ATCC7469-M sample, followed by CICC6117-M, CICC21790-M and CICC22704-M, and CK samples. After fermentation, there was a significant increase in the levels of 2, 3-butanedione, acetoin, and 2-heptanone in this study. These three compounds are key aroma contributors in fermented dairy products and positively enhance the product’s aroma, characterized by their intense fragrance and creamy notes [18]. Interestingly, among the fermented samples, CICC22704-M exhibited the highest concentration of 2, 3-butanedione (approximately 305.95 μg/L), whereas CICC21790-M, ATCC7469-M, and CICC6117-M showed higher levels of acetoin (above 450 μg/L). Previous studies have reported that acetoin acts as the precursor for 2, 3-butanedione, and that the enzymes acetolactate synthase and α-acetolactate decarboxylase play a critical role in the synthesis of 2, 3-butanedione [9]. Therefore, CICC22704 may exhibit higher activities of these two enzymes compared with other strains, thereby promoting the synthesis of 2, 3-butanedione and reducing the content of acetion. Moreover, a higher acetoin content was reported in the fermented milk with *Lacticaseibacillus rhamnosus* WH.FH-19 [45]. For 2-heptanone, higher contents (above 202 μg/L) were observed in the CICC21790-M, ATCC7469-M, and CICC6117-M samples, which might be attributed to the activity of strains CICC21790, ATCC7469, and CICC6117 to promote the biotransformation of linoleic acid [46]. Additionally, compared with the CK sample, the fermented samples exhibited increased levels of 2-butanone and decreased levels of 2, 3-pentanedione and 2-pentanone. Previous studies have indicated that LABs are capable of producing 2-butanone during fermentation [8]. According to the Flavornet database, 2, 3-pentanedione is characterized by a creamy and buttery flavor profile, whereas 2-pentanone exhibits sweet and fruity flavor notes. These two compounds were not detected in soymilk [8,9], which may be attributed to the saccharified chestnut solution used for fermentation. The reduction in these two compounds may be associated with the metabolic activities of different strains, necessitating further investigation.

Esters generated during LAB fermentation are primarily derived from fatty acids. Those produced from short-chain fatty acids emit a fruity aroma, whereas those derived from long-chain fatty acids exhibit a fatty flavor [47]. Compared to the CK sample, CICC22704-M exhibited a significantly higher concentration of total esters. Similar findings have been reported in soy milk, peanut milk, and chickpea milk fermented by *Limosilactobacillus fermentum* GD01 [48]. This phenomenon may be explained by the strong acid-producing capability of CICC22704, which is subsequently converted into esters through further reactions [32]. However, ATCC7469-M, CICC21790-M, and CICC6117-M showed markedly lower levels of total esters, which might be caused by a reduction in hexyl acetate (Table 3). This compound was not observed in soymilk [8,9] and may be derived from saccharified chestnut solution. The decreased level of hexyl acetate was observed in the pea-based fermented beverages through *Lactobacillus delbrueckii subsp. bulgaricus* and *Streptococcus thermophilus* fermentation [49].

Acid compounds are closely associated with lipid decomposition, microbial fermentation, and other metabolic pathways [50]. During soymilk fermentation, LAB can produce a variety of organic acids, such as acetic acid and propionic acid, which impart a soft sour taste to the product [32]. Additionally, this process can effectively eliminate off-odors caused by certain raw materials. In this study, compared with the CK sample, both total and individual acid compounds were significantly increased in the fermented samples, particularly in the CICC22704-M and ATCC7469-M samples. Acetic acid, one of the two detected acid compounds, accounted for a substantial proportion of the acids in the samples. It is produced via the hetero-lactic fermentation pathway utilized by LAB, and both CICC22704 and ATCC7469 are hetero-fermentative strains [19]. Consequently, the production of acetic acid by CICC22704 and ATCC7469 was significantly higher than that by CICC21790 and CICC6117.

Furan compounds were commonly detected in both soymilk and fermented soymilk [8,9], as identified in this study. Compared with the CK sample, a higher concentration of furan compounds was observed in the fermented samples. Among the four furan compounds detected, 2-pentylfuran was the most abundant in both unfermented and fermented samples. It is primarily formed through the reaction of singlet oxygen with linoleic acid [38], while the presence of light and air can further influence its production [51]. Following fermentation, the concentration of 2-pentylfuran in the fermented samples was slightly elevated compared with that in the CK sample, a finding that aligns with previous research [9]. Interestingly, the marked increase in furan compounds observed exclusively in the CICC22704-M sample was attributed to 2-furanmethanol, which can be explained by the inherent ability of CICC22704 to produce this caramel-like compound, thereby enhancing the flavor profile [52].

The sulfur-containing compounds in soymilk are likely formed via the thermal degradation of lysine, cysteine, and methionine at high temperatures [51,53]. Compared with the CK sample, an elevated concentration of sulfur-containing compounds was observed in the CICC22704-M and ATCC7469-M samples, which validates the findings of the e-nose and indicates that these two strains are indeed good producers of organic sulfide compounds. Notably, CICC22704-M contained a higher concentration of dimethyl disulfide, 2-methylthiophene, and methionol, which suggests that CICC22704 may possess a strong capacity to enhance amino acid metabolism associated with the production of sulfur-containing compounds. Styrene exhibits a distinctive odor, reminiscent of resin and flowers [54], which may originate from the saccharified chestnut solution. After fermentation, only CICC22704 notably increased the styrene content; however, there are currently no reports on styrene production through LAB fermentation. Therefore, further research is necessary to clarify this phenomenon. For the unknown compounds, the fermented samples exhibited a higher concentration of unknown 6, while only CICC22704-M contained a higher level of unknown 2.

#### 3.3.3. PCA Analysis of Volatiles in Plant-Based Milks

To better understand the differences in volatile compositions between unfermented and fermented plant-based milks, PCA was carried out using the concentrations of all volatiles detected via HS-GC-IMS as variables. In the PCA plot, samples with similar aroma profiles tend to cluster closely together, potentially even overlapping, whereas samples with distinct aroma characteristics are clearly separated from one another [28]. As shown in Figure 3, PC1 and PC2 account for 50.4% and 29.7% of the total variance, respectively, thereby representing the overall sample distribution. The results revealed a clear distinction between unfermented and fermented samples, which can be attributed to the significant contributions of various compounds with high loadings in the positive direction of PC1 and negative direction of PC2. In the CK sample, the high loading values of AH1, AH2, AH4, AH5, AH6, AH7, AH8, AH9, AH10, AO12, AO16, K3, K4, E8, UC1, and UC6 in the negative direction of PC1 were identified as the primary contributors. For the fermented samples, the high loading values of SC1, SC2, SC3, A1, AO1, AO2, AO10, AO11, E2, E3, E5, E7, F2, K1, EL1, and UC2 in the positive direction of PC1 were determined as the main contributors to the CICC22704-M sample. However, the compounds A2, AO3, AO5, AO6, AO7, AO9, F1, F4, K2, K5, K6, K8, AH3, UC3, and UC4 with high loading values in the negative direction of PC2 were associated with the CICC21790-M, CICC6117-M, and ATCC7469-M samples.

#### 3.3.4. Screen the Key Volatile Flavor Compounds of the Plant-Based Milks

To better understand the differences in aroma characteristics between unfermented and fermented plant-based milks, the variable importance in projection (VIP) values and ROAV were combined to identify the key flavor compounds and assess their contributions to the overall aroma profile of these samples. The aroma descriptions and aroma types of volatile compounds with VIP ≥ 1.0 and ROAV ≥ 1.0 in these samples are shown in Table 4, and visualization of aroma contribution diagrams for these compounds is shown in Figure 4. In the CK sample, the compounds with higher contributions were predominantly aldehydes (e.g., hexanal with ROAV = 100.00 and heptanal with ROAV = 35.80), followed by ketones (e.g., 2, 3-butanedione with ROAV = 33.18), and finally furans (e.g., 2-pentylfuran with ROAV = 34.80). These compounds primarily impart Green, Creamy, and Fruity aroma characteristics to the overall aroma profile of the CK sample (Figure 4). To date, no prior studies have reported the significant contribution of ketones to soymilk [8,9]. Previous studies on the roasting of chestnuts have documented the presence of ketones [55,56]. Consequently, we speculate that ketones may originate from the roasting process of chestnuts. As expected, the contribution of aldehydes to the flavor decreased significantly in the fermented samples, particularly in the CICC22704-M sample. Thus, CICC22704 fermentation may notably reduce the Green aroma characteristic compared to the other three strains (Figure 4). It was reported that this aroma characteristic is an undesired odor in the fermented plant-based milks [8]. Simultaneously, increased ROAVs of 1-pentanol, 2, 3-butanedione, and 2-pentylfuran were observed in the fermented samples, which likely enhanced the Pungent, Creamy, and Fruity aroma characteristics (Figure 4). Notably, CICC21790, ATCC7469, and CICC6117 fermentation might result in stronger intensities of these three aroma characteristics compared to CICC22704 fermentation. Creamy and Fruity aroma characteristics have been reported as desired odors for the fermented plant-based milks [8] while the Pungent aroma characteristic is considered an undesired odor. Additionally, the Fatty aroma characteristic, which is related to an undesired odor of fermented plan-based milk [8], showed an increased tendency only in CICC21790 fermentation. It was significantly reduced by ATCC7469, CICC6117, and CICC22704 fermentation, especially by CICC22704.

## 4. Conclusions

In this study, different fermented plant-based milks composed of chestnut and soybean were developed using *Lactiplantibacillus plantarum* CICC21790, *Lacticaseibacillus casei* CICC6117, *Lacticaseibacillus rhamnosus* ATCC7469, and *Limosilactobacillus fermentum* CICC22704. The fermentative characteristics of the four LAB strains and the aroma characteristics of the resulting plant-based milks were analyzed. The four LAB strains exhibit superior adaptability to the fermentation environment of a saccharified chestnut solution and soymilk mixture. Using an e-nose and GC-IMS analysis, significant differences in flavor profiles were observed between the plant-based milk before and after fermentation. The Creamy aroma characteristics of unfermented plant-based milk may be attributed to the roasting process of chestnuts. All four lactic acid bacteria strains’ fermentation could weaken the Green aroma characteristic, especially CICC22704. Fermentation with CICC21790, ATCC7469, and CICC6117 might result in stronger intensities of Pungent, Creamy, and Fruity aroma characteristics. For the Fatty aroma characteristic, it was enhanced by CICC21790 fermentation but diminished by ATCC7469, CICC6117, and CICC22704 fermentation. The results obtained in this study can provide a theoretical basis for the development of plant-based food products as well as the industrialization and further processing of chestnuts. Future investigations integrating metabolomics and transcriptomics are anticipated to provide a more comprehensive understanding of how lactic acid bacteria modulate flavor.

## Figures and Tables

**Figure 1 foods-14-02511-f001:**
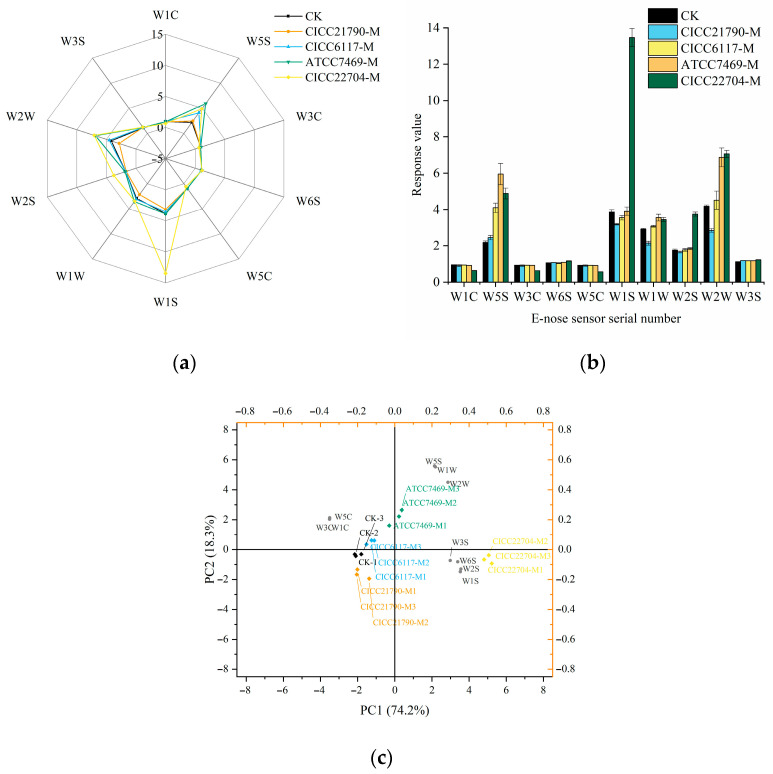
E-nose analysis of plant-based milks. (**a**) Radar map; (**b**) Column chart; (**c**) Biplot of PCA for plant-base milks. Note: The fermented plant-based milks obtained using *Lactiplantibacillus plantarum* CICC21790, *Lacticaseibacillus casei* CICC6117, *Lacticaseibacillus rhamnosus* ATCC7469, and *Limosilactobacillus fermentum* CICC22704 strains were designated as CICC21790-M, CICC6117-M, ATCC7469-M, and CICC22704-M, respectively, while the unfermented plant-based milk was designated as CK.

**Figure 2 foods-14-02511-f002:**
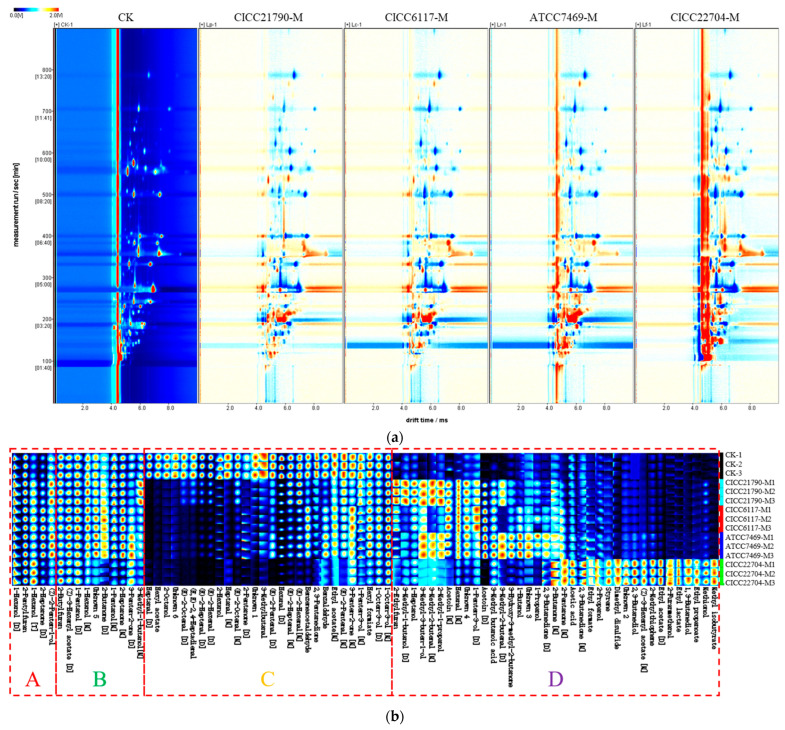
Volatiles fingerprints analysis of the volatile compounds in plant-based milks. (**a**) The 2D difference comparison topographic plots of the volatile compounds. (**b**) Gallery plot fingerprints of the volatile compounds. Note: The fermented plant-based milks obtained using *Lactiplantibacillus plantarum* CICC21790, *Lacticaseibacillus casei* CICC6117, *Lacticaseibacillus rhamnosus* ATCC7469, and *Limosilactobacillus fermentum* CICC22704 strains were designated as CICC21790-M, CICC6117-M, ATCC7469-M, and CICC22704-M, respectively, while the unfermented plant-based milk was designated as CK. Certain substances have the abbreviations [M], [D], and [T], referring to the monomer, dimer and trimer of the same element.

**Figure 3 foods-14-02511-f003:**
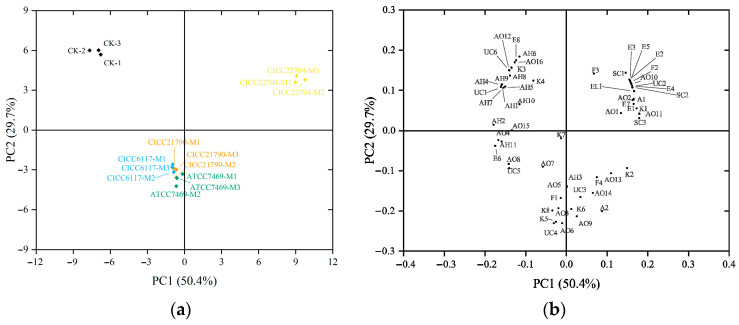
PCA analysis of volatiles in plant-based milks. (**a**) PCA score plot graph; (**b**) PCA loading plot graph. Note: The AH, AO, E, K, A, F, SC, EL, UC represent the aldehydes, alcohols, esters, ketones, acids, furans, sulfur compounds, else, unknown compounds, respectively. The fermented plant-based milks obtained using *Lactiplantibacillus plantarum* CICC21790, *Lacticaseibacillus casei* CICC6117, *Lacticaseibacillus rhamnosus* ATCC7469, and *Limosilactobacillus fermentum* CICC22704 strains were designated as CICC21790-M, CICC6117-M, ATCC7469-M, and CICC22704-M, respectively, while the unfermented plant-based milk was designated as CK.

**Figure 4 foods-14-02511-f004:**
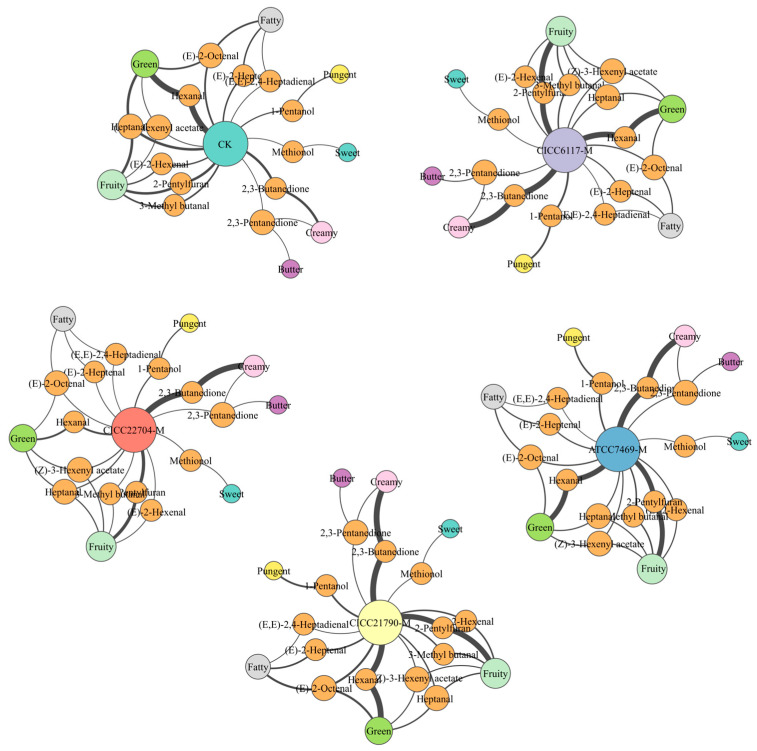
Aroma characteristic of the key flavor compounds in plant-based milks. Note: The fermented plant-based milks obtained using *Lactiplantibacillus plantarum* CICC21790, *Lacticaseibacillus casei* CICC6117, *Lacticaseibacillus rhamnosus* ATCC7469, and *Limosilactobacillus fermentum* CICC22704 strains were designated as CICC21790-M, CICC6117-M, ATCC7469-M, and CICC22704-M, respectively, while the unfermented plant-based milk was designated as CK.

**Table 1 foods-14-02511-t001:** Performance description of E-nose sensors.

Array No.	Sensor Name	Performance Description
1	W1C	Sensitive to aromatic benzene
2	W5S	Very sensitive to nitrogen oxides, especially negative to nitrogen oxides
3	W3C	Ammonia, sensitive to aromatic components
4	W6S	Mainly selective to hydrides
5	W5C	Short-chain alkanes, aromatic compounds sensitive
6	W1S	Sensitive to methyls
7	W1W	Sensitive to inorganic sulfides and terpenes
8	W2S	Sensitive to alcohols, aldehydes, and ketones
9	W2W	Aromatic ingredients, sensitive to organic sulfur compounds
10	W3S	Sensitive to long-chain alkanes

**Table 2 foods-14-02511-t002:** Fermentative characteristics of CICC21790, CICC6117, ATCC7469, and CICC22704.

Samples	Viable Cell Count (lg CFU mL^−1^)	Soluble Solid Content °Brix (%)	Reducing Sugar Content (g/L)	pH Value	Titratable Acidity (°T)
CK	0 ^e^	8.5 ± 0.5 ^a^	37.65 ± 0.65 ^a^	6.27 ± 0.02 ^a^	17.53 ± 0.86 ^c^
CICC21790-M	7.67 ± 0.02 ^d^	6.0 ± 0.5 ^b^	29.19 ± 0.26 ^b^	3.83 ± 0.01 ^c^	85.50 ± 0.50 ^a^
CICC6117-M	8.57 ± 0.01 ^a^	6.0 ± 0.5 ^b^	29.61 ± 0.67 ^b^	3.87 ± 0.07 ^c^	83.76 ± 0.86 ^b^
ATCC7469-M	8.11 ± 0.03 ^b^	6.0 ± 0.5 ^b^	30.25 ± 0.67 ^b^	3.88 ± 0.03 ^c^	82.47 ± 1.72 ^b^
CICC22704-M	7.95 ± 0.05 ^c^	6.0 ± 0.5 ^b^	29.19 ± 0.19 ^b^	4.07 ± 0.05 ^b^	87.77 ± 2.76 ^a^

Note: Values followed by different letters in the same column are significantly different (*p* < 0.05) by Duncan test. The fermented plant-based milks obtained using *Lactiplantibacillus plantarum* CICC21790, *Lacticaseibacillus casei* CICC6117, *Lacticaseibacillus rhamnosus* ATCC7469, and *Limosilactobacillus fermentum* CICC22704 strains were designated as CICC21790-M, CICC6117-M, ATCC7469-M, and CICC22704-M, respectively, while the unfermented plant-based milk was designated as CK.

**Table 3 foods-14-02511-t003:** The volatile compounds identified by GC-IMS in plant-based milks.

No.	Compounds	CAS#	Formula	MW	RI	Rt [second]	Dt [a.u.]	Concentration (μg/L)				
								CK	CICC21790-M	CICC6117-M	ATCC7469-M	CICC22704-M
	Aldehydes							1927.08 ± 90.06 ^a^	864.62 ± 34.50 ^b^	621.54 ± 30.19 ^d^	732.19 ± 30.39 ^c^	267.24 ± 16.20 ^e^
**AH1**	3-Methylbutanal	C590863	C_5_H_10_O	86.1	652.5	166.00	1.19	55.96 ± 2.48 ^a^	19.26 ± 0.91 ^bc^	17.90 ± 0.73 ^c^	20.71 ± 1.34 ^b^	6.07 ± 0.55 ^d^
	(E)-2-Pentenal [M]	C1576870	C_5_H_8_O	84.1	754.0	232.16	1.10	70.56 ± 4.94 ^a^	58.32 ± 2.65 ^b^	47.61 ± 2.25 ^c^	46.38 ± 1.98 ^c^	9.90 ± 0.53 ^d^
	(E)-2-Pentenal [D]	C1576870	C_5_H_8_O	84.1	753.9	232.05	1.36	107.63 ± 6.09 ^a^	53.23 ± 2.15 ^c^	42.09 ± 1.68 ^d^	69.04 ± 3.23 ^b^	3.92 ± 0.36 ^e^
**AH2**	(E)-2-Pentenal							178.19 ± 11.03 ^a^	111.55 ± 4.79 ^b^	89.70 ± 3.92 ^c^	115.42 ± 5.17 ^b^	13.82 ± 0.89 ^d^
	3-Methyl-2-butenal [M]	C107868	C_5_H_8_O	84.1	785.3	259.99	1.09	4.36 ± 0.51 ^c^	13.35 ± 0.50 ^a^	2.97 ± 0.14 ^d^	9.56 ± 0.34 ^b^	4.33 ± 0.16 ^c^
	3-Methyl-2-butenal [D]	C107868	C_5_H_8_O	84.1	782.2	257.11	1.36	0.41 ± 0.02 ^c^	4.14 ± 0.13 ^a^	0.55 ± 0.05 ^c^	4.31 ± 0.40 ^a^	1.87 ± 0.16 ^b^
**AH3**	3-Methyl-2-butenal							4.77 ± 0.50 ^d^	17.49 ± 0.57 ^a^	3.52 ± 0.11 ^e^	13.87 ± 0.71 ^b^	6.20 ± 0.32 ^c^
	Hexanal [M]	C66251	C_6_H_12_O	100.2	795.7	270.05	1.27	13.90 ± 0.43 ^d^	90.36 ± 3.59 ^a^	93.62 ± 3.67 ^a^	82.51 ± 3.26 ^b^	45.60 ± 2.50 ^c^
	Hexanal [D]	C66251	C_6_H_12_O	100.2	798.9	273.16	1.56	681.21 ± 30.72 ^a^	187.55 ± 6.69 ^bc^	172.23 ± 11.05 ^c^	202.59 ± 8.12 ^b^	75.04 ± 5.59 ^d^
**AH4**	Hexanal							695.11 ± 31.00 ^a^	277.92 ± 10.15 ^b^	265.85 ± 14.66 ^b^	285.10 ± 11.37 ^b^	120.64 ± 7.91 ^c^
	(E)-2-Hexenal [M]	C6728263	C_6_H_10_O	98.1	854.4	334.60	1.18	127.54 ± 5.28 ^a^	91.96 ± 4.36 ^b^	51.85 ± 2.33 ^d^	61.94 ± 2.75 ^c^	19.22 ± 1.64 ^e^
	(E)-2-Hexenal [D]	C6728263	C_6_H_10_O	98.1	853.3	333.19	1.52	210.55 ± 9.66 ^a^	77.96 ± 3.16 ^b^	23.48 ± 1.04 ^d^	41.72 ± 1.40 ^c^	11.08 ± 0.86 ^e^
**AH5**	(E)-2-Hexenal							338.09 ± 14.94 ^a^	169.92 ± 7.49 ^b^	75.33 ± 3.37 ^d^	103.67 ± 4.13 ^c^	30.29 ± 2.49 ^e^
	Heptanal [M]	C111717	C_7_H_14_O	114.2	902.9	400.43	1.35	79.48 ± 4.57 ^a^	19.26 ± 0.97 ^b^	15.51 ± 0.58 ^bc^	13.75 ± 0.54 ^c^	17.90 ± 0.95 ^b^
	Heptanal [D]	C111717	C_7_H_14_O	114.2	902.3	399.45	1.70	86.43 ± 4.15 ^a^	4.88 ± 0.40 ^c^	3.64 ± 0.23 ^c^	3.77 ± 0.31 ^c^	11.70 ± 0.41 ^b^
**AH6**	Heptanal							165.92 ± 8.43 ^a^	24.14 ± 1.34 ^bc^	19.14 ± 0.80 ^c^	17.52 ± 0.84 ^c^	29.61 ± 0.98 ^b^
	(E)-2-Heptenal [M]	C18829555	C_7_H_12_O	112.2	960.0	500.61	1.25	163.53 ± 4.32 ^a^	105.56 ± 4.44 ^b^	57.55 ± 2.70 ^d^	69.91 ± 3.01 ^c^	22.05 ± 1.72 ^e^
	(E)-2-Heptenal [D]	C18829555	C_7_H_12_O	112.2	960.5	501.58	1.66	136.43 ± 6.13 ^a^	46.18 ± 1.32 ^b^	14.75 ± 0.67 ^d^	21.97 ± 0.84 ^c^	6.74 ± 0.71 ^e^
**AH7**	(E)-2-Heptenal							299.96 ± 10.35 ^a^	151.75 ± 5.72 ^b^	72.30 ± 3.33 ^d^	91.88 ± 3.69 ^c^	28.79 ± 2.39 ^e^
**AH8**	(E,E)-2,4-Heptadienal	C4313035	C_7_H_10_O	110.2	1022.4	622.79	1.19	29.99 ± 3.21 ^a^	12.16 ± 0.95 ^b^	3.71 ± 0.12 ^c^	4.37 ± 0.18 ^c^	2.32 ± 0.22 ^c^
	(E)-2-Octenal [M]	C2548870	C_8_H_14_O	126.2	1065.7	706.62	1.83	79.09 ± 3.93 ^a^	45.27 ± 1.40 ^b^	18.28 ± 1.50 ^d^	22.43 ± 1.02 ^c^	6.77 ± 0.25 ^e^
	(E)-2-Octenal [D]	C2548870	C_8_H_14_O	126.2	1064.4	703.92	1.32	16.51 ± 1.20 ^a^	4.84 ± 0.43 ^b^	1.73 ± 0.10 ^c^	2.03 ± 0.16 ^c^	1.51 ± 0.11 ^c^
**AH9**	(E)-2-Octenal							995.61 ± 5.12 ^a^	50.11 ± 1.79 ^b^	20.01 ± 1.59 ^c^	24.46 ± 1.17 ^c^	8.28 ± 0.34 ^d^
**AH10**	Benzaldehyde	C100527	C_7_H_6_O	106.1	964.4	509.44	1.15	24.41 ± 1.83 ^a^	11.44 ± 0.48 ^e^	18.17 ± 1.07 ^c^	22.02 ± 1.06 ^b^	13.92 ± 0.69 ^d^
**AH11**	Benzeneacetaldehyde	C122781	C_8_H_8_O	120.2	1040.1	655.88	1.26	39.08 ± 1.75 ^a^	18.89 ± 0.81 ^d^	35.92 ± 1.49 ^b^	33.18 ± 1.15 ^c^	7.30 ± 0.24 ^e^
	Alcohols							1780.70 ± 77.20 ^a^	1923.85 ± 87.86 ^a^	1886.90 ± 113.78 ^a^	1870.64 ± 86.35 ^a^	1923.42 ± 85.14 ^a^
**AO1**	1-Propanol	C71238	C_3_H_8_O	60.1	531.4	119.13	1.12	42.19 ± 7.61 ^c^	79.96 ± 19.43 ^ab^	31.49 ± 22.77 ^c^	55.27 ± 19.86 ^bc^	108.67 ± 5.37 ^a^
**AO2**	2-Propanol	C67630	C_3_H_8_O	60.1	534.5	120.14	1.09	38.25 ± 1.71 ^c^	38.83 ± 2.03 ^c^	42.57 ± 10.90 ^bc^	51.10 ± 3.68 ^b^	87.73 ± 5.69 ^a^
**AO3**	2-Methyl-1-propanol	C78831	C_4_H_10_O	74.1	626.9	154.77	1.17	6.66 ± 0.29 ^d^	19.40 ± 0.56 ^a^	9.75 ± 0.56 ^c^	16.32 ± 0.95 ^b^	6.91 ± 0.38 ^d^
	1-Penten-3-ol [M]	C616251	C_5_H_10_O	86.1	686.4	182.21	1.34	101.20 ± 4.01 ^a^	73.88 ± 2.68 ^b^	62.34 ± 4.53 ^c^	59.23 ± 2.31 ^c^	39.51 ± 1.26 ^d^
	1-Penten-3-ol [D]	C616251	C_5_H_10_O	86.1	682.3	170.13	0.94	10.38 ± 1.10 ^e^	29.80 ± 1.11 ^b^	39.15 ± 0.49 ^a^	26.49 ± 1.08 ^c^	24.49 ± 1.10 ^d^
**AO4**	1-Penten-3-ol							111.58 ± 5.09 ^a^	103.68 ± 3.79 ^b^	101.49 ± 4.97 ^b^	85.72 ± 3.21 ^c^	64.00 ± 2.35 ^d^
**AO5**	1-Butanol	C71363	C_4_H_10_O	74.1	688.8	183.44	1.37	65.07 ± 5.07 ^d^	59.05 ± 3.84 ^d^	140.35 ± 3.21 ^b^	214.65 ± 8.34 ^a^	81.44 ± 3.76 ^c^
**AO6**	3-Methyl-3-buten-1-ol	C763326	C_5_H_10_O	86.1	732.0	214.45	1.17	2.91 ± 0.49 ^e^	17.05 ± 0.69 ^a^	12.64 ± 0.56 ^c^	14.14 ± 0.66 ^b^	4.67 ± 0.14 ^d^
	3-Methyl-1-butanol [M]	C123513	C_5_H_12_O	88.1	736.0	217.55	1.25	39.59 ± 2.19 ^b^	54.42 ± 2.07 ^a^	33.46 ± 2.17 ^c^	34.92 ± 1.42 ^c^	28.16 ± 1.79 ^d^
	3-Methyl-1-butanol [D]	C123513	C_5_H_12_O	88.1	733.9	215.87	1.49	3.56 ± 0.13 ^d^	11.14 ± 0.21 ^a^	4.13 ± 0.26 ^d^	6.77 ± 0.64 ^b^	5.99 ± 0.06 ^c^
**AO7**	3-Methyl-1-butanol							43.15 ± 2.10 ^b^	65.56 ± 1.90 ^a^	37.59 ± 2.37 ^c^	41.68 ± 2.06 ^b^	34.16 ± 1.78 ^c^
	1-Pentanol [M]	C71410	C_5_H_12_O	88.1	768.9	245.02	1.25	150.59 ± 5.99 ^a^	150.45 ± 5.82 ^a^	152.54 ± 6.43 ^a^	132.92 ± 6.86 ^b^	97.22 ± 4.31 ^c^
	1-Pentanol [D]	C71410	C_5_H_12_O	88.1	765.8	242.25	1.53	114.62 ± 2.99 ^ab^	117.77 ± 5.09 ^ab^	122.19 ± 5.13 ^a^	111.33 ± 5.19 ^b^	113.89 ± 4.33 ^ab^
**AO8**	1-Pentanol							265.21 ± 8.97 ^a^	268.22 ± 10.91 ^a^	274.73 ± 11.56 ^a^	244.25 ± 11.98 ^b^	211.11 ± 8.41 ^c^
**AO9**	(Z)-2-Penten-1-ol	C1576950	C_5_H_10_O	86.1	774.8	250.30	0.94	13.62 ± 0.76 ^e^	22.46 ± 0.77 ^b^	19.64 ± 1.53 ^c^	25.99 ± 0.76 ^a^	17.23 ± 0.60 ^d^
**AO10**	1,3-Butanediol	C107880	C_4_H_10_O_2_	90.1	782.7	257.52	1.14	13.35 ± 0.59 ^b^	13.28 ± 0.26 ^b^	12.18 ± 0.30 ^b^	11.61 ± 1.28 ^b^	80.97 ± 3.10 ^a^
**AO11**	2,3-Butanediol	C513859	C_4_H_10_O_2_	90.1	795.7	270.05	1.37	7.58 ± 0.21 ^d^	28.24 ± 1.25 ^c^	28.40 ± 1.87 ^c^	36.52 ± 3.22 ^b^	96.73 ± 5.66 ^a^
**AO12**	2-Hexanol	C626937	C_6_H_14_O	102.2	808.3	282.72	1.27	138.22 ± 3.98 ^a^	29.44 ± 1.28 ^b^	27.26 ± 2.48 ^b^	28.79 ± 0.93 ^b^	12.32 ± 1.03 ^c^
	1-Hexanol [M]	C111273	C_6_H_14_O	102.2	874.5	360.05	1.64	187.62 ± 5.42 ^a^	190.42 ± 7.58 ^a^	187.19 ± 7.85 ^a^	179.46 ± 7.90 ^a^	131.67 ± 6.22 ^b^
	1-Hexanol [D]	C111273	C_6_H_14_O	102.2	872.9	357.93	1.99	363.63 ± 15.51 ^d^	544.42 ± 22.06 ^a^	505.25 ± 22.98 ^b^	425.23 ± 18.37 ^c^	561.33 ± 26.24 ^a^
	1-Hexanol [T]	C111273	C_6_H_14_O	102.2	881.3	369.05	1.33	17.97 ± 1.17 ^d^	35.11 ± 1.36 ^b^	34.44 ± 2.22 ^b^	29.34 ± 1.61 ^c^	44.01 ± 1.75 ^a^
**AO13**	1-Hexanol							569.22 ± 21.81 ^c^	769.95 ± 30.60 ^a^	726.88 ± 32.48 ^a^	634.03 ± 27.28 ^b^	737.01 ± 33.20 ^a^
**AO14**	1-Heptanol	C111706	C_7_H_16_O	116.2	979.3	539.87	1.40	9.25 ± 0.59 ^e^	27.61 ± 1.39 ^a^	20.70 ± 0.89 ^b^	15.46 ± 0.58 ^d^	18.25 ± 0.83 ^c^
	1-Octen-3-ol [M]	C3391864	C_8_H_16_O	128.2	988.0	558.53	1.16	297.81 ± 14.91 ^a^	282.89 ± 12.09 ^ab^	293.20 ± 12.93 ^ab^	289.34 ± 11.61 ^ab^	268.82 ± 11.59 ^b^
	1-Octen-3-ol [D]	C3391864	C_8_H_16_O	128.2	987.0	556.47	1.59	106.81 ± 3.06 ^a^	89.42 ± 4.29 ^c^	99.43 ± 4.99 ^b^	96.95 ± 4.58 ^b^	85.68 ± 2.92 ^c^
**AO15**	1-Octen-3-ol							404.62 ± 17.94 ^a^	372.31 ± 16.30 ^bc^	392.63 ± 17.82 ^ab^	386.29 ± 16.18 ^ab^	354.50 ± 14.34 ^c^
**AO16**	2-Octanol	C123966	C_8_H_18_O	130.2	990.4	563.83	1.45	49.81 ± 2.97 ^a^	8.80 ± 0.31 ^b^	8.58 ± 0.83 ^b^	8.83 ± 0.58 ^b^	7.73 ± 0.87 ^b^
	Esters							367.68 ± 16.28 ^b^	272.76 ± 11.71 ^c^	292.89 ± 12.16 ^c^	306.44 ± 12.64 ^c^	1206.97 ± 52.35 ^a^
**E1**	Ethyl formate	C109944	C_3_H_6_O_2_	74.1	609.3	147.49	1.21	20.89 ± 1.05 ^d^	20.01 ± 1.49 ^d^	27.17 ± 3.50 ^c^	35.19 ± 2.65 ^b^	61.72 ± 2.52 ^a^
	Ethyl acetate [M]	C141786	C_4_H_8_O_2_	88.1	614.2	149.48	1.10	36.26 ± 4.30 ^a^	30.04 ± 1.11 ^b^	31.53 ± 1.79 ^b^	27.73 ± 1.63 ^b^	18.74 ± 1.17 ^c^
	Ethyl acetate [D]	C141786	C_4_H_8_O_2_	88.1	607.1	146.60	1.34	52.34 ± 1.36 ^b^	36.15 ± 2.16 ^c^	37.30 ± 2.52 ^c^	43.19 ± 1.78 ^bc^	335.50 ± 12.84 ^a^
**E2**	Ethyl acetate							88.60 ± 4.98 ^b^	66.19 ± 2.79 ^c^	68.84 ± 3.78 ^c^	70.92 ± 2.97 ^c^	354.24 ± 13.83 ^a^
**E3**	Methyl isobutyrate	C547637	C_5_H_10_O_2_	102.1	654.5	166.93	1.13	27.69 ± 1.38 ^b^	16.51 ± 1.31 ^c^	18.22 ± 1.16 ^c^	17.14 ± 0.59 ^c^	208.57 ± 5.89 ^a^
**E4**	Ethyl propanoate	C105373	C_5_H_10_O_2_	102.1	718.8	204.39	1.14	21.37 ± 0.24 ^b^	27.46 ± 0.59 ^b^	24.65 ± 0.94 ^b^	17.96 ± 0.68 ^b^	170.96 ± 17.84 ^a^
**E5**	Ethyl lactate	C97643	C_5_H_10_O_3_	118.1	818.4	293.38	1.13	46.76 ± 2.70 ^b^	36.09 ± 2.05 ^b^	37.48 ± 0.70 ^b^	36.90 ± 2.42 ^b^	259.95 ± 11.71 ^a^
**E6**	Hexyl formiate	C629334	C_7_H_14_O_2_	130.2	869.0	352.82	1.33	53.84 ± 2.20 ^a^	45.03 ± 1.89 ^b^	47.58 ± 2.30 ^b^	48.56 ± 2.38 ^b^	30.36 ± 1.63 ^c^
	(Z)-3-Hexenyl acetate [M]	C3681718	C_8_H_14_O_2_	142.2	991.9	567.26	1.04	23.04 ± 1.03 ^c^	25.52 ± 0.88 ^c^	27.50 ± 0.70 ^c^	36.57 ± 3.68 ^b^	85.03 ± 3.88 ^a^
	(Z)-3-Hexenyl acetate [D]	C3681718	C_8_H_14_O_2_	142.2	991.9	567.28	1.31	34.63 ± 1.68 ^a^	30.40 ± 2.01 ^b^	36.97 ± 1.53 ^a^	38.00 ± 1.84 ^a^	30.90 ± 1.94 ^b^
**E7**	(Z)-3-Hexenyl acetate							57.67 ± 2.59 ^d^	55.92 ± 2.07 ^d^	64.46 ± 2.22 ^c^	74.57 ± 3.99 ^b^	115.93 ± 5.64 ^a^
**E8**	Hexyl acetate	C142927	C_8_H_16_O_2_	144.2	1012.6	605.27	1.42	50.87 ± 1.89 ^a^	5.55 ± 0.20 ^b^	4.47 ± 0.61 ^b^	5.19 ± 0.49 ^b^	5.24 ± 0.21 ^b^
	Ketones							616.98 ± 34.80 ^e^	1055.51 ± 56.94 ^c^	1357.44 ± 41.23 ^b^	1546.95 ± 64.18 ^a^	761.90 ± 37.21 ^d^
	2, 3-Butanedione [M]	C431038	C_4_H_6_O_2_	86.1	591.3	140.39	1.18	113.24 ± 14.27 ^bc^	129.03 ± 12.61 ^b^	101.39 ± 2.57 ^bc^	110.95 ± 8.73 ^c^	266.70 ± 17.25 ^a^
	2, 3-Butanedione [D]	C431038	C_4_H_6_O_2_	86.1	587.7	139.00	1.05	20.01 ± 0.87 ^d^	32.88 ± 2.69 ^c^	59.51 ± 8.01 ^b^	75.37 ± 4.57 ^a^	39.25 ± 2.52 ^c^
**K1**	2, 3-Butanedione							133.24 ± 15.08 ^c^	161.91 ± 15.25 ^b^	160.90 ± 7.67 ^b^	186.32 ± 13.27 ^b^	305.95 ± 18.03 ^a^
	2-Butanone [M]	C78933	C_4_H_8_O	72.1	594.8	141.75	1.25	35.05 ± 3.41 ^d^	53.63 ± 4.97 ^b^	49.87 ± 3.36 ^bc^	43.48 ± 2.33 ^c^	72.74 ± 3.79 ^a^
	2-Butanone [D]	C78933	C_4_H_8_O	72.1	594.4	141.57	1.07	19.50 ± 0.44 ^c^	23.84 ± 0.81 ^b^	28.67 ± 1.95 ^a^	26.84 ± 1.03 ^a^	12.12 ± 0.67 ^d^
**K2**	2-Butanone							54.54 ± 3.08 ^c^	77.47 ± 5.77 ^ab^	78.55 ± 5.30 ^ab^	70.32 ± 3.33 ^b^	84.86 ± 4.46 ^a^
**K3**	2,3-Pentanedione	C600146	C_5_H_8_O_2_	100.1	695.6	187.97	1.20	68.99 ± 2.65 ^a^	35.47 ± 1.80 ^b^	31.21 ± 1.73 ^c^	29.30 ± 1.42 ^c^	28.29 ± 1.48 ^c^
	2-Pentanone [M]	C107879	C_5_H_10_O	86.1	696.3	188.49	1.13	3.33 ± 0.22 ^e^	10.13 ± 0.59 ^d^	17.09 ± 0.13 ^c^	22.30 ± 2.36 ^b^	39.29 ± 1.88 ^a^
	2-Pentanone [D]	C107879	C_5_H_10_O	86.1	699.6	190.69	1.38	70.85 ± 5.56 ^a^	18.72 ± 0.91 ^d^	29.34 ± 2.70 ^c^	40.13 ± 1.48 ^b^	8.51 ± 0.17 ^e^
**K4**	2-Pentanone							74.18 ± 5.34 ^a^	28.85 ± 1.39 ^d^	46.43 ± 2.74 ^c^	62.44 ± 3.76 ^b^	47.79 ± 1.89 ^c^
	Acetoin [M]	C513860	C_4_H_8_O_2_	88.1	716.2	202.52	1.33	11.13 ± 7.23 ^c^	71.88 ± 2.71 ^a^	69.41 ± 4.88 ^a^	55.02 ± 2.56 ^b^	15.23 ± 0.34 ^c^
	Acetoin [D]	C513860	C_4_H_8_O_2_	88.1	724.2	208.43	1.07	32.88 ± 4.36 ^d^	378.71 ± 19.02 ^c^	549.57 ± 8.13 ^b^	667.33 ± 30.16 ^a^	39.73 ± 14.76 ^d^
**K5**	Acetoin							44.01 ± 11.55 ^d^	450.59 ± 21.40 ^c^	618.98 ± 11.43 ^b^	722.35 ± 32.60 ^a^	54.96 ± 14.81 ^d^
**K6**	3-Hydroxy-3-methyl-2-butanone	C115220	C_5_H_10_O_2_	102.1	732.9	215.10	1.40	12.53 ± 0.14 ^e^	42.23 ± 2.71 ^b^	37.84 ± 0.89 ^c^	77.23 ± 3.86 ^a^	24.93 ± 1.48 ^d^
	3-Penten-2-one [M]	C625332	C_5_H_8_O	84.1	738.4	219.42	1.33	40.53 ± 4.03 ^a^	26.85 ± 1.14 ^c^	38.32 ± 1.83 ^ab^	34.97 ± 1.70 ^b^	24.31 ± 1.60 ^c^
	3-Penten-2-one [D]	C625332	C_5_H_8_O	84.1	738.8	219.74	1.07	35.10 ± 1.24 ^d^	29.49 ± 1.41 ^e^	41.86 ± 0.42 ^c^	55.40 ± 2.62 ^a^	48.87 ± 1.95 ^b^
**K7**	3-Penten-2-one							75.63 ± 5.27 ^bc^	56.34 ± 2.54 ^d^	80.17 ± 1.52 ^b^	90.36 ± 4.32 ^a^	73.18 ± 3.47 ^c^
	2-Heptanone [M]	C110430	C_7_H_14_O	114.2	892.9	385.02	1.27	78.08 ± 3.49 ^b^	79.21 ± 2.74 ^b^	105.39 ± 4.41 ^a^	105.61 ± 4.71 ^a^	44.89 ± 2.16 ^c^
	2-Heptanone [D]	C110430	C_7_H_14_O	114.2	892.7	384.79	1.64	75.77 ± 2.97 ^d^	123.43 ± 4.85 ^b^	197.97 ± 7.36 ^a^	203.02 ± 8.13 ^a^	97.05 ± 3.45 ^c^
**K8**	2-Heptanone							153.84 ± 6.44 ^c^	202.64 ± 7.45 ^b^	303.35 ± 11.74 ^a^	308.62 ± 12.82 ^a^	141.94 ± 5.45 ^c^
	Acids							188.00 ± 6.13 ^d^	222.20 ± 14.08 ^cd^	231.21 ± 16.28 ^c^	290.10 ± 28.21 ^b^	450.68 ± 24.05 ^a^
**A1**	Acetic acid	C64197	C_2_H_4_O_2_	60.1	657.7	168.42	1.04	183.57 ± 4.46 ^c^	190.03 ± 10.91 ^c^	197.39 ± 15.57 ^c^	247.71 ± 30.15 ^b^	421.59 ± 21.89 ^a^
**A2**	3-Methyl butanoic acid	C503742	C_5_H_10_O_2_	102.1	840.0	317.38	1.23	4.43 ± 2.85 ^c^	32.17 ± 4.35 ^b^	33.82 ± 0.73 ^b^	42.39 ± 3.98 ^a^	29.09 ± 2.21 ^b^
	Furans							362.46 ± 16.50 ^c^	405.20 ± 16.10 ^b^	397.95 ± 16.53 ^b^	394.04 ± 19.06 ^bc^	473.50 ± 22.27 ^a^
**F1**	2-Ethylfuran	C3208160	C_6_H_8_O	96.1	706.6	195.63	1.07	5.88 ± 2.16 ^d^	31.97 ± 1.21 a	22.11 ± 0.86 b	10.86 ± 0.44 c	7.57 ± 1.05 d
**F2**	2-Furanmethanol	C98000	C_5_H_6_O_2_	98.1	851.6	331.10	1.13	15.97 ± 1.40 ^b^	14.81 ± 0.86 ^b^	11.65 ± 0.14 ^b^	13.08 ± 1.10 ^b^	99.15 ± 7.89 ^a^
**F3**	2-Butylfuran	C4466244	C_8_H_12_O	124.2	894.9	388.09	1.18	18.04 ± 0.70 ^ab^	18.17 ± 0.91 ^a^	16.38 ± 1.05 ^bc^	15.17 ± 0.72 ^c^	19.47 ± 1.13 ^a^
**F4**	2-Pentylfuran	C3777693	C_9_H_14_O	138.2	997.6	579.33	1.24	322.56 ± 14.65 ^b^	340.25 ± 13.52 ^ab^	347.82 ± 14.89 ^ab^	354.93 ± 16.95 ^a^	347.31 ± 15.22 ^ab^
	Sulfur compounds							93.81 ± 2.46 ^c^	107.65 ± 4.76 ^bc^	100.75 ± 4.89 ^bc^	118.99 ± 11.63 ^b^	421.29 ± 17.49 ^a^
**SC1**	Dimethyl disulfide	C624920	C_2_H_6_S_2_	94.2	752.1	230.59	1.13	20.97 ± 0.92 ^b^	12.08 ± 0.30 ^c^	11.10 ± 0.49 ^c^	11.16 ± 0.52 ^c^	72.78 ± 3.42 ^a^
**SC2**	2-Methylthiophene	C554143	C_5_H_6_S	98.2	782.4	257.26	1.05	61.43 ± 1.44 ^c^	55.99 ± 2.49 ^c^	61.88 ± 3.36 ^c^	77.40 ± 10.54 ^b^	266.50 ± 11.60 ^a^
**SC3**	Methionol	C505102	C_4_H_10_OS	106.2	977.3	535.66	1.09	11.42 ± 1.13 ^d^	39.58 ± 1.98 ^b^	27.77 ± 1.46 ^c^	30.44 ± 1.39 ^c^	82.01 ± 3.24 ^a^
	Else							173.13 ± 10.45 ^b^	127.41 ± 5.04 ^d^	136.97 ± 7.41 ^d^	155.28 ± 6.38 ^c^	202.92 ± 7.57 ^a^
**EL1**	Styrene	C100425	C_8_H_8_	104.2	884.9	373.99	1.41	20.95 ± 0.65 ^c^	22.71 ± 0.75 ^bc^	23.20 ± 0.64 ^bc^	25.61 ± 2.39 ^b^	67.78 ± 3.34 ^a^
**UC1**	Unknown 1							19.37 ± 1.15 ^a^	8.12 ± 0.68 ^b^	6.54 ± 0.27 ^c^	6.34 ± 0.38 ^c^	1.95 ± 0.13 ^d^
**UC2**	Unknown 2							24.86 ± 2.24 ^b^	23.85 ± 1.08 ^b^	24.97 ± 1.65 ^b^	27.27 ± 1.74 ^b^	78.06 ± 2.74 ^a^
**UC3**	Unknown 3							12.04 ± 1.23 ^d^	17.58 ± 1.03 ^c^	23.95 ± 1.15 ^b^	36.19 ± 1.76 ^a^	19.31 ± 0.83 ^c^
**UC4**	Unknown 4							8.24 ± 0.37 ^b^	18.04 ± 0.87 ^a^	16.45 ± 2.30 ^a^	17.46 ± 0.45 ^a^	8.10 ± 0.75 ^b^
**UC5**	Unknown 5							28.85 ± 1.69 ^ab^	27.36 ± 1.25 ^b^	30.14 ± 1.71 ^a^	28.59 ± 1.28 ^ab^	22.78 ± 1.01 ^c^
**UC6**	Unknown 6							58.83 ± 3.94 ^a^	9.75 ± 0.26 ^c^	11.73 ± 1.16 ^bc^	13.82 ± 0.74 ^b^	4.94 ± 0.50 ^d^
**Total**								5509.85 ± 245.75 ^a^	4979.21 ± 225.77 ^b^	5025.64 ± 241.47 ^b^	5414.63 ± 238.47 ^ab^	5707.93 ± 242.02 ^a^

Note: In the context of the provided information, the abbreviations represent the following; MW: Molecular Weight; RI: Retention Index; Dt: Relative Migration Time. The AH, AO, E, K, A, F, SC, EL, UC represent the aldehydes, alcohols, esters, ketones, acids, furans, sulfur compounds, else, unknown compounds, respectively. The fermented plant-based milks obtained using *Lactiplantibacillus plantarum* CICC21790, *Lacticaseibacillus casei* CICC6117, *Lacticaseibacillus rhamnosus* ATCC7469, and *Limosilactobacillus fermentum* CICC22704 strains were designated as CICC21790-M, CICC6117-M, ATCC7469-M, and CICC22704-M, respectively, while the unfermented plant-based milk was designated as CK. The letters “^a^”, “^b^”, “^c^”, “^d^”, and “^e^” represent significant differences (*p* < 0.05) in volatile compounds among unfermented and fermented plant-based milks.

**Table 4 foods-14-02511-t004:** Volatiles with VIPs ≥ 1.0 and ROAV ≥1.0 in plant-based milks identified by HS-SPME-GC-IMS.

No.	Compounds	VIP	Odor Threshold (μg/kg)	Aroma Description	Aroma Type	ROAV				
						CK	CICC21790-M	CICC6117-M	ATCC7469-M	CICC22704-M
**1**	3-Methyl butanal	1.16	2 [42]	apple, peach [42]	Fruity	18.11	15.46	14.46	14.45	2.58
**2**	Hexanal	1.22	4.5 [42]	Grassy [42]	Green	100.00	99.17	95.46	88.41	22.78
**3**	(E)-2-Hexenal	1.31	17 [42]	Fruity [42]	Fruity	12.87	16.05	7.16	8.51	1.51
**4**	Heptanal	1.34	3 [42]	fruity, grassy [42]	Fruity; Green	35.80	12.92	10.31	8.15	8.39
**5**	(E)-2-Heptenal	1.34	13 [32]	woody, fatty [32]	Fatty	14.94	18.74	8.99	9.86	1.88
**6**	(E,E)-2,4-Heptadienal	1.54	15 [32]	fried, nut, fatty [57]	Fatty	1.29	1.30	0.40	0.41	0.13
**7**	(E)-2-Octenal	1.41	3 [57]	fresh, cucumber, fatty [57]	Green; Fatty	20.63	26.82	10.78	11.38	2.35
**8**	1-Pentanol	1.03	20 [42]	Spicy, wine [42]	Pungent	8.58	21.54	22.20	17.04	8.97
**9**	(Z)-3-Hexenyl acetate	1.21	13 ^a^	green, banana ^b^	Green; Fruity	2.87	6.91	8.01	8.00	7.58
**10**	2, 3-Butanedione	1.18	2.6 ^a^	Cream [58]	Creamy	33.18	100.00	100.00	100.00	100.00
**11**	2, 3-Pentanedione	1.51	20 [28]	Cream, butter ^b^	Creamy; Butter	2.23	2.85	2.52	2.04	1.20
**12**	2-Pentylfuran	1.15	6 [32]	Fruity [32]	Fruity	34.80	91.06	93.67	82.55	49.19
**13**	Methionol	1.10	36 ^a^	sweet, potato ^b^	Sweet	0.21	1.77	1.25	1.18	1.94

Note: ^a^ Odor threshold referenced the ‘Compilations of odor threshold values in air, water and other media’. ^b^ Odor descriptions were cited from www.flavornet.org. The fermented plant-based milks obtained using *Lactiplantibacillus plantarum* CICC21790, *Lacticaseibacillus casei* CICC6117, *Lacticaseibacillus rhamnosus* ATCC7469, and *Limosilactobacillus fermentum* CICC22704 strains were designated as CICC21790-M, CICC6117-M, ATCC7469-M, and CICC22704-M, respectively, while the unfermented plant-based milk was designated as CK.

## Data Availability

The original contributions presented in the study are included in the article, further inquiries can be directed to the corresponding author.

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
