# Peer review of "Fermented Plant-Based Milks Based on Chestnut and Soybean: Comprehensive Evaluation of Fermentation Characteristics and Aroma Profiles Using Four Lactic Acid Bacteria Strains"

_foods, 2025, doi:10.3390/foods14142511_

Round 1
Reviewer 1 Report
Comments and Suggestions for Authors
This study presents a research paper based on the analysis of fermentation characteristics and flavor components of fermented soymilk mixed with saccharified chestnut solution, and fermented using various Lactobacillus (LAB) strains.
It is judged to be systematically well written in terms of the paper structure, research methodology, and research results.
However, some parts require revision, so I am leaving a comment below.
[Introduction]
Line 49-51: This study placed a high emphasis on flavor analysis. Therefore, please simplify or delete contents such as lines 55-58, and describe specifically what flavor compounds can be formed by the use of carbohydrates during soymilk fermentation.
Line 58-60: “Hebei Province .... production [14].” This sentence seems unnecessary.
Line 67-69: Are hexanal and 1-octen-3-ol the off-flavor-causing substances in soy milk?
Line 77-90:
1) There is no need for unnecessary length about the methodology of this study.
2) Rather, I would like you to emphasize more the necessity of adding chestnuts to soy milk, which was briefly mentioned in lines 35-45.
Line 91: In “LAB strains (Lp, Lc, Lr, Lf)”, please state the full name first before using the abbreviation.
[Mateinals and Methods]
Line 101, 116, 122, 143: Please provide information about the equipment used in this study.
[2.2. Preparation of plant-based milk] section: The authors mixed a saccharified chestnut solution and sterilized soymilk in a 2:3 ratio. However, is there any prior research literature they referenced to establish this ratio?
[Results and Discussion]
[3.1. Fermentative characteristics analysis of LAB strains] section: There are significant differences between LAB strain samples in the ‘Viable cell count’, ‘pH value’, and ‘Titratable acidity (°T)’ sections. Could you elaborate a bit more on this?
Line 222-227: Please describe in detail the aromas produced by the flavor components mentioned by the authors.
Line 239-241: When performing PCA analysis on volatile flavor compounds derived from electronic nose analysis results, the ratio of PC1 and PC2 indicates which of the x-axis (PC1) and y-axis (PC2) has a higher contribution.
[Volatiles fingerprints of plant-based milks] section: The resolution of the x-axis and y-axis in Fig. 2a and 2b is low, making them difficult to read. Please improve this.
Line 281-283: Please describe in detail the aromas produced by the compounds mentioned by the authors.
Line 383-384: Please describe additional research cases similar to those discussed by the authors.
Line 393-394: Please add appropriate references to the authors' discussion.
[3.3.3. PCA analysis of volatiles in plant-based milks] section: Fig. 3b needs to be adjusted position.
Line 519-521 “The results obtained in ... alternative to animal-based yogurt”: I recommend that you delete this part.
Comments on the Quality of English LanguageThe overall English quality is understandable, and the conventions of paper expression are generally well-written.
However, there is room for improvement in some grammatical errors and vocabulary choices.
Author Response
Comments 1: Line 49-51: This study placed a high emphasis on flavor analysis. Therefore, please simplify or delete contents such as lines 55-58, and describe specifically what flavor compounds can be formed by the use of carbohydrates during soymilk fermentation.
Response 1: Thank you for your professional advice. In the revised manuscript, we have added detailed information about which flavor compounds can be affected by the use of carbohydrates during soymilk fermentation (in lines 51-56).
Details are as follows:
As reported previously, during soymilk fermentation, the addition of some carbohydrates (such as glucose or lactose) can modulate the glycolysis/gluconeogenesis and pyruvate metabolic pathways, thereby promoting the production of metabolites including 2, 3-butanedione, acetoin, acetaldehyde, and acetic acid [8]. These flavor substances can improve the undesirable flavor of plant-based fermented products to some extent [9].
Comments 2: Line 58-60: “Hebei Province .... production [14].” This sentence seems unnecessary.
Response 2: Thanks for your careful review. We have deleted the sentence in the revised manuscript.
Comments 3: Line 67-69: Are hexanal and 1-octen-3-ol the off-flavor-causing substances in soy milk?
Response 3: Yes, hexanal and 1-octen-3-ol have been shown to enhance the beany flavor in soy milk, which is generally regarded as an undesirable off-flavor. This has been confirmed by the following references.
References:
Zheng, Y.; Fei, Y.T.; Yang, Y.; Jin, Z.K.; Yu, B.N.; Li, L. A potential flavor culture: Lactobacillus harbinensis M1 improves the organoleptic quality of fermented soymilk by high production of 2, 3-butanedione and acetoin. Food Microbiol. 2020, 91, 103540.
Yi, C.P.; Li, Y.S.; Zhu, H.; Liu, Y.L.; Quan, K. Effect of Lactobacillus plantarum fermentation on the volatile flavors of mung beans. LWT-Food Sci. Technol. 2021, 146, 111434.
Yu, G.W.; Hua, Y.F.; Zhang, C.M,; Li, X.F.; Kong, X.Z.; Chen, Y.M. Characterization of the volatile organic compounds and sensory properties of fermented soymilks as affected by carbohydrates and starter cultures. LWT. 2024, 202, 116264.
Comments 4: Line 77-90:
1) There is no need for unnecessary length about the methodology of this study.
2) Rather, I would like you to emphasize more the necessity of adding chestnuts to soy milk, which was briefly mentioned in lines 35-45.
Response 4: Thank you for your kind suggestions. The necessity of adding chestnuts to soymilk was discussed in lines 56-68.
Here are the details:
As is well known, chestnuts are highly favored by consumers due to their unique flavor, texture and nutritional richness [10, 11]. Starch constitutes the primary component of chestnuts, accounting for approximately 70%–80% on a dry weight basis [12]. Following saccharification, chestnuts exhibit increased sugar content, which can serve as an energy source for the growth and metabolism of LAB [13]. Moreover, the oligosaccharides produced during the saccharification process have been shown to effectively promote the proliferation of LAB [12]. Additionally, chestnuts are rich in polysaccharides, flavonoids, polyphenols, and other bioactive substances, which confer numerous physiological bene-fits, including antioxidant, hypoglycemic, and anti-inflammatory properties [14]. There-fore, we attempted to blend soymilk with saccharified chestnut solution to produce a fermented plant-based milk. This approach enables us to leverage the high protein content of soymilk as well as the abundant sugar content and distinctive aroma of chestnuts.
Comments 5: Line 91: In “LAB strains (Lp, Lc, Lr, Lf)”, please state the full name first before using the abbreviation.
Response 5: Thank you. We have added the full names of lactic acid bacterial strains into this sentence (in lines 98-99).
Comments 6: Line 101, 116, 122, 143: Please provide information about the equipment used in this study.
Response 6: Thank you for your professional suggestion. We have added the detailed information of the equipment.
The details are as follows:
lines 108-109: Special shelling machine (Small and simple type, Jiangsu Xinda Machinery Manufacturing Co., Ltd., China)
lines 125-126: Magnetic stirring oil bath pot (Four-hole variable temperature and variable stirring digital display model,Hunan Qianyan Technology Co., Ltd., China)
lines 132-133: Clean bench (OptiClean-1300, Likang Precision Technology Co., Ltd., China)
line 155: pH meter (FE28, Mettler-Toledo Instruments Co., Ltd., China)
Comments 7: [2.2. Preparation of plant-based milk] section: The authors mixed a saccharified chestnut solution and sterilized soymilk in a 2:3 ratio. However, is there any prior research literature they referenced to establish this ratio?
Response 7: Thank you for comments. Before the fermentation, we screened the ratio of saccharified chestnut solution and sterilized soymilk through pre-experiment. It was found that the plant-based fermented milk showed favorable curdling properties and flavor at a ratio of 1:1.5. Therefore, we used this ratio in subsequent experiments.
Comments 8: [3.1. Fermentative characteristics analysis of LAB strains] section: There are significant differences between LAB strain samples in the ‘Viable cell count’, ‘pH value’, and ‘Titratable acidity (°T)’ sections. Could you elaborate a bit more on this?
Response 8: In the revised manuscript, we have described the significant differences among the fermented samples in viable cell count, pH value, and titratable acidity (°T) (in lines 217-222, and 226-227). Thank you again for the kind comments.
Comments 9: Line 222-227: Please describe in detail the aromas produced by the flavor components mentioned by the authors.
Response 9: Thank you for your suggestion. Due to the fact that the E-nose signal output typically reflects a composite response to multiple volatile compounds, differentiation can only be achieved at the category level rather than for individual volatile. Therefore, the response values obtained from the e-nose sensor are insufficient to characterize the aroma profiles of volatile compounds.
Reference:
Ma, Y.H.; Wang, Y.Y.; Li, J.; Wang, B.; Li, M.; Ma, T.Z.; Jiang, Y.M.; Zhang, B. Volatile organic compound dynamics in Ugni Blanc and Vidal wines during fermentation in the Hexi Corridor (China): insights from E-nose, GC-MS, GC-IMS, and multivariate statistical models, LWT. 2025, 217, 117440.
Comments 10: Line 239-241: When performing PCA analysis on volatile flavor compounds derived from electronic nose analysis results, the ratio of PC1 and PC2 indicates which of the x-axis (PC1) and y-axis (PC2) has a higher contribution.
Response 10: Thank you for your valuable suggestions. We have revised this sentence accordingly (in lines 259-260).
Comments 11: [Volatiles fingerprints of plant-based milks] section: The resolution of the x-axis and y-axis in Fig. 2a and 2b is low, making them difficult to read. Please improve this.
Response 11: Thank you for your suggestion. We have enhanced the resolution of the figure.
Comments 12: Line 281-283: Please describe in detail the aromas produced by the compounds mentioned by the authors.
Response 12: Thank you for your suggestion. In this section, the peak intensities in the volatile compound fingerprint were analyzed to offer a visual representation of the variations in volatile compound concentrations before and after fermentation. Whether these volatile compounds have an impact on the aroma characteristics of the sample needs to be evaluated through a comprehensive analysis using both VIP and ROAV. A more comprehensive analysis regarding the aroma profiles of volatile compounds can be found in Section 3.3.2 and Section 3.3.4. Thank you again for your kind suggestion.
Reference:
Yi, C.P.; Li, Y.S.; Zhu, H.; Liu, Y.L.; Quan, K. Effect of Lactobacillus plantarum fermentation on the volatile flavors of mung beans. LWT-Food Sci. Technol. 2021, 146, 111434.
Zheng, A.R.; , Wei, C.K.; Wang, M.S.; Ju, N.; Fan, M. Characterization of the key flavor compounds in cream cheese by GC-MS, GC-IMS, sensory analysis and multivariable statistics. Current research in food science. 2024, 8, 100772.
Comments 13: Line 383-384: Please describe additional research cases similar to those discussed by the authors.
Response 13: Thank you for pointing out. We have revised the statement accordingly in the manuscript (in lines 404-407).
Here are the details:
These two compounds were not detected in soymilk [8, 9], which may be attributed to the saccharified chestnut solution used for fermentation. The reduction in these two com-pounds may be associated with the metabolic activities of different strains, necessitating further investigation.
Comments 14: Line 393-394: Please add appropriate references to the authors' discussion.
Response 14: Thank you for careful reading. This statement is derived from the experimental results presented in Table 3. We deeply apologize for any confusion caused by the ambiguity in wording, which may have hindered your understanding. This sentence has now been revised accordingly and the relevant reference has been added in the revised manuscript (in lines 415-420).
The details are as follows:
However, ATCC7469-M, CICC21790-M, and CICC6117-M showed markedly lower levels of total esters, which might be caused by a reduction in hexyl acetate (Table 3). This compound was not observed in soymilk [8, 9] and may be derived from saccharified chestnut solution. The decreased level of hexyl acetate was observed in the pea-based fermented beverages through Lactobacillus delbrueckii subsp. bulgaricus and Streptococcus thermophilus fermentation [48].
Reference:
Alinovi, M.; Bancalari, E.; Monica, S.; Del Vecchio, L.; Cirlini, M.; Chiavaro, E. Bot, F. Tailoring the physico-chemical properties and VOCs of pea-based fermented beverages through Lactobacillus delbrueckii subsp. bulgaricus and Streptococcus thermophilus fermentation. Food Res Int. 2025. 209, 116250.
Comments 15: [3.3.3. PCA analysis of volatiles in plant-based milks] section: Fig. 3b needs to be adjusted position.
Response 15: We apologize for our careless mistake. In the revised manuscript, we have adjusted the position of Fig.3b. Thanks again for your kind comments.
Comments 16: Line 519-521 “The results obtained in ... alternative to animal-based yogurt”: I recommend that you delete this part.
Response 16: Thank you for your kind suggestion. We have deleted the sentence of “The results obtained in ... alternative to animal-based yogurt”.
Comments 17: The overall English quality is understandable, and the conventions of paper expression are generally well-written. However, there is room for improvement in some grammatical errors and vocabulary choices.
Response 17: We sincerely thank you for careful reading. Upon receiving your valuable feedback, I promptly undertook a comprehensive review and revision of the manuscript. The entire document has undergone thorough linguistic refinement to enhance the accuracy, fluency, and clarity of expression.
Reviewer 2 Report
Comments and Suggestions for Authors
General/major comments
In this paper, the authors evaluate the fermentation characteristics and aromatic profiles of four strains of lactic acid bacteria cultivated on plant-base milk. This is an important analytical work which necessarily remains quite descriptive but which nevertheless provides interesting elements regarding the potential of lactic bacteria in the production of plant-based milks.
The manuscript is generally well written, the methods are well developed and the results do not call for major comments. It is an important work that provides relevant information regarding the links lactic acid bacteria and aromatic profiles of the fermented products.
Minor comments
Throughout the manuscript, use the up-to-date Lactobacillus classification (i.e. Lactobacillus plantarum is Lactiplantibacillus plantarum)
L16 : Change the word "kinds" which is not used in systematic nomenclature to "strains"
L25: The sentence "The increased concentration..." is not clear at all. Please modify it.
L142: Why was this dosage method used? What is the reliability and accuracy of this DNS method? Furthermore, the method's detailed protocol, or at least a reference, is missing.
As I mentioned in my report, I consider this to be an interesting study aimed at characterizing the fermentation profiles of four strains of lactic acid bacteria. The subject seems relevant to me and in line with the theme of the FOODS journal. This provides new data on the diversity of volatile compounds produced by the four strains of lactic acid bacteria selected for this study. The limitation of the study is that these results are not necessarily transposable to other strains of the same bacterial genera, as the production of volatile compounds may be strain-dependent. The conclusion is relevant in light of the results presented, and the bibliography is appropriate. Given the large amount of data presented, the tables and figures are sometimes difficult to understand; the authors may wish to review the presentation of certain data, particularly the extremely busy Figure 2.
Author Response
Comments 1: Throughout the manuscript, use the up-to-date Lactobacillus classification (i.e. Lactobacillus plantarum is Lactiplantibacillus plantarum)
Response 1: Thank you for your professional suggestion. The nomenclature of lactic acid bacteria has been updated in accordance with the latest taxonomic standards, and the entire manuscript has been thoroughly revised to ensure consistency.
Comments 2: L16: Change the word "kinds" which is not used in systematic nomenclature to "strains"
Response 2: Thank you for pointing out. We have replaced the word "kinds" with "strains" in the revised manuscript (in line 16).
Comments 3: L25: The sentence "The increased concentration..." is not clear at all. Please modify it.
Response 3: Thank you for your professional suggestion. We have revised the manuscript according to your suggestion (in lines 26-28).
The details are as follows:
The increased concentrations of alcohols (e.g., 1-pentanol), ketones (e.g., 2, 3-butanedione) and furan compounds (e.g., 2-pentylfuran) in fermented plant-based milks enhanced Pungent, Creamy, and Fruity aroma characteristics, respectively.
Comments 4: L142: Why was this dosage method used? What is the reliability and accuracy of this DNS method? Furthermore, the method's detailed protocol, or at least a reference, is missing.
Response 4: Thank you very much. The DNS method exhibits high sensitivity towards these commonly occurring reducing sugars. We have included the relevant reference in the revised manuscript (in line 154).
Reference:
Hao, J.; Xu, H.N.; Yan, P.F.; Yang, M.Y.; Mintah, B.K.; Gao, X.L.; Zhang, R. Effect of ultrasound-assisted fermentation on physicochemical properties and volatile flavor compounds of Chinese rice wine. Food physics. 2024, 1, 100006.
Comments 5: As I mentioned in my report, I consider this to be an interesting study aimed at characterizing the fermentation profiles of four strains of lactic acid bacteria. The subject seems relevant to me and in line with the theme of the FOODS journal. This provides new data on the diversity of volatile compounds produced by the four strains of lactic acid bacteria selected for this study. The limitation of the study is that these results are not necessarily transposable to other strains of the same bacterial genera, as the production of volatile compounds may be strain-dependent. The conclusion is relevant in light of the results presented, and the bibliography is appropriate. Given the large amount of data presented, the tables and figures are sometimes difficult to understand; the authors may wish to review the presentation of certain data, particularly the extremely busy Figure 2.
Response 5: We sincerely appreciate your thorough and insightful review. Your comments have been invaluable in helping us enhance the quality of the manuscript. In accordance with your suggestions, we have removed certain figure, including Figure 2a, to improve the clarity and readability of the manuscript. Thank you again for your professional and friendly comments.
Round 2
Reviewer 1 Report
Comments and Suggestions for Authors
I have confirmed that the manuscript has been revised to reflect the comments I mentioned earlier.
- Line 72-75: As in the comment I mentioned earlier, please describe that hexanal and 1-octen-3-ol are off-flavor substances derived from soy milk.
- [2.2. Preparation of plant-based milk] section: Please describe that the authors conducted a preliminary experiment to establish the mixing ratio of saccharified chestnut solution and sterilized soymilk.
Author Response
Comment 1: Line 72-75: As in the comment I mentioned earlier, please describe that hexanal and 1-octen-3-ol are off-flavor substances derived from soy milk.
Response 1: Thank you for your professional suggestion. We have revised this sentence accordingly (lines 72-76).
Here are the details:
Moreover, fermentation using Streptococcus thermophilus, Lactiplantibacillus plantarum, Bifidobacterium, and Lacticaseibacillus casei individually could markedly reduce the concentrations of hexanal and 1-octen-3-ol, which are derived from soymilk [16-19] and lead to the typical soybean off-flavor [9].
Comment 2: [2.2. Preparation of plant-based milk] section: Please describe that the authors conducted a preliminary experiment to establish the mixing ratio of saccharified chestnut solution and sterilized soymilk.
Response 2: Thanks for your careful review. We have described the mixing ratio of the saccharified chestnut solution and sterilized soymilk based on preliminary experiments in this section (lines 132-137).
Here are the details:
Finally, on a clean bench (OptiClean-1300, Likang Precision Technology Co., Ltd., China), the saccharified chestnut solution and sterilized soymilk were mixed in a 2:3 (m/m) ratio under sterile conditions to prepare the plant-based milk. It should be noted that the ratio of the saccharified chestnut solution to sterilized soymilk was determined through preliminary trials by evaluating the curdling properties and flavor of the resulting fermented plant-based milk.